Co-inhibition of adenosine 2b receptor and programmed death-ligand 1 promotes the recruitment and cytotoxicity of natural killer cells in oral squamous cell carcinoma

Wang Bing 1
Wang Tao 1
Yang Chengzhe 1
Nan Zhaodi 2
Ai Dan 2
Wang Xin 3
Wang Huayang 4
Qu Xun quxun@sdu.edu.cn 2
Wei Fengcai fengcaiwei@outlook.com 1
1 Department of Oral and Maxillofacial Surgery, Qilu Hospital, Cheeloo College of Medicine, Shandong University & Institute of Stomatology, Cheeloo College of Medicine, Shandong University , Jinan , China
2 Institute of Basic Medical Sciences, Qilu Hospital, Cheeloo College of Medicine, Shandong University , Jinan , China
3 Department of Pathology, School of Basic Medical Sciences, Cheeloo College of Medicine, Shandong University , Jinan , China
4 Department of Clinical Laboratory, Qilu Hospital, Cheeloo College of Medicine, Shandong University , Jinan , China
Dong Peixin
Electronic publication date: 2023 Aug 30
Publication date: 2023
Volume: 11
Electronic Location ID: e15922
Received 2023 Jun 1; Accepted 2023 Jul 28
Copyright: ©2023 Wang et al.
Copyright year: 2023
Copyright holder: Wang et al.
License: This is an open access article distributed under the terms of the Creative Commons Attribution License, which permits unrestricted use, distribution, reproduction and adaptation in any medium and for any purpose provided that it is properly attributed. For attribution, the original author(s), title, publication source (PeerJ) and either DOI or URL of the article must be cited.
License URL: https://creativecommons.org/licenses/by/4.0/

Keywords: Oral squamous cell carcinoma, Checkpoint inhibitor, PD-L1, A2BR, NK cell

Funding: The National Natural Science Foundation of China (NSFC) No.81772879 This work was supported by the National Natural Science Foundation of China (NSFC) (No.81772879). The funders had no role in study design, data collection and analysis, decision to publish, or preparation of the manuscript.

==============================
Adenosine promotes anti-tumor immune responses by modulating the functions of T-cells and natural killer (NK) cells in the tumor microenvironment; however, the role of adenosine receptors in the progression of oral squamous cell carcinoma (OSCC) and its effects on immune checkpoint therapy remain unclear. In this study, we obtained the tumor tissues from 80 OSCC patients admitted at the Shandong University Qilu Hospital between February 2014 and December 2016. Thereafter, we detected the expression of adenosine 2b receptor (A2BR) and programmed death-ligand 1 (PD-L1) using immunohistochemical staining and analyzed the association between their expression in different regions of the tumor tissues, such as tumor nest, border, and paracancer stroma. To determine the role of A2BR in PD-L1 expression, CAL-27 (an OSCC cell line) was treated with BAY60-6583 (an A2BR agonist), and PD-L1 expression was determined using western blot and flow cytometry. Furthermore, CAL-27 was treated with a nuclear transcription factor-kappa B (NF-κ B) inhibitor, PDTC, to determine whether A2BR regulates PD-L1 expression via the NF-κ B signaling pathway. Additionally, a transwell assay was performed to verify the effect of A2BR and PD-L1 on NK cell recruitment. The results of our study demonstrated that A2BR and PD-L1 are co-expressed in OSCC. Moreover, treatment with BAY60-6583 induced PD-L1 expression in the CAL-27 cells, which was partially reduced in cells pretreated with PDTC, suggesting that A2BR agonists induce PD-L1 expression via the induction of the NF-κ B signaling pathway. Furthermore, high A2BR expression in OSCC was associated with lower infiltration of NK cells. Additionally, our results demonstrated that treatment with MRS-1706 (an A2BR inverse agonist) and/or CD274 (a PD-L1-neutralizing antibody) promoted NK cell recruitment and cytotoxicity against OSCC cells. Altogether, our findings highlight the synergistic effect of co-inhibition of A2BR and PD-L1 in the treatment of OSCC via the modulation of NK cell recruitment and cytotoxicity.

Introduction

Head and neck cancers (HNC) are the sixth most common type of cancer worldwide, and squamous cell carcinomas account for approximately 90% of all HNC cases. Oral squamous cell carcinoma (OSCC) is the most common type of oral malignancy (Wang et al., 2021). Despite substantial progress in conventional OSCC treatments, such as surgery, radiotherapy, and chemotherapy, there has been no significant improvement in the OSCC survival rate in the past few decades (Chai, Lim & Cheong, 2020; Li et al., 2023). Recently, immune checkpoint inhibitors received increasing attention as novel anticancer agents, and targeting immune checkpoint proteins, such as programmed cell death protein 1 (PD-1) and PD-1 ligand 1 (PD-L1), has shown remarkable clinical efficacy in multiple malignancies at low cytotoxicity (Meliante et al., 2022; Togo et al., 2020). Moreover, several clinical studies have demonstrated that checkpoint blockade immunotherapy exhibits significant clinical effects and leads to a significant increase in the survival rate. However, due to tumor heterogeneity less than 20% of the OSCC patients benefit from checkpoint blockade immunotherapy (Chiu, Ou & Tan, 2022; Trumet et al., 2023). Therefore, two major challenges for immunotherapy are to determine the underlying mechanisms associated with ICI resistance and to decrease drug cytotoxicity. A potential approach to determine the molecular mechanisms associated with ICI resistance is to identify the novel molecular targets which enhance anti-tumor immunity in OSCC patients.

The tumor microenvironment (TME) includes tumor cells, local stromal cells, infiltrated immune cells, and other components, which together constitute a dynamic interacting network that modulates the biological behavior of tumor cells. Moreover, the high heterogeneity of TME has a substantial impact on the therapeutic effects of anti-cancer treatments. Immune cell infiltration is a characteristic feature of TME. Currently, most of the immunomodulatory strategies for anti-cancer treatments focus on enhancing T-cell response (Shimu et al., 2023). Moreover, abnormal expression of MHC I on tumor cells and upregulation of PD-L1 are associated with poor prognosis in Hodgkin lymphoma patients, thus suggesting that immune response is independent of T-cells, inhibited by PD-1, and rescued by PD-L1 blockade (Mulder et al., 2022). Natural killer (NK) cells have cytolytic activity and are involved in the immune monitoring of tumor cells and virus-infected cells (Peng et al., 2021). Studies have shown that NK cells generate multiple immune responses during tumorigenesis, compared with the types of immune cells, such as T cells, NK cells exhibit continual antitumor immune response (Yu, 2023).

Adenosine is a critical intermediate for the synthesis of adenosine triphosphate (ATP), adenine, adenylate, and vidarabine. Adenosine receptors (ARs) are members of the G protein-coupled receptors superfamily and play important roles in energy metabolism and immunity (Luo et al., 2022; Nakamura et al., 2020). They are classified into four receptor types: A1R, A2AR, A2BR, and A3R (Saini et al., 2022; Zhong, Peng & Zeng, 2022), among which A2BR is expressed in various cell types and requires higher adenosine concentration for activation (Sun, Wang & Hao, 2022). Moreover, under A2BR deficiency, a mouse ischemic heart model showed an increased number of immune cells, including B cells, NK cells, and CD4 T-cells. Additionally, A2BR plays an important role in the proliferation, differentiation, and apoptosis of cancer cells (Guieu et al., 2021; Augustin et al., 2022). Generation of purine nucleosides in TME effectively inhibits T-cell and NK cell activities, studies have found that A2AR and A3R play critical roles in tumor immune responses (Zahavi & Hodge, 2023; Strickland et al., 2023). Therefore, ARs have been used as therapeutic targets for novel drug development against cancers. Moreover, studies have found that A2BR is highly expressed in oral cancer tissues and is closely associated with tumor progression, suggesting that A2BR might be a potential target for drug development. However, the role of A2BR in anti-tumor immunity against OSCC remains unclear.

In this study, we explored the correlation between A2BR and PD-L1 expression and NK cell infiltration in OSCC tissues. Additionally, we conducted in vitro experiments to determine the mechanisms by which co-inhibition of A2BR and PD-L1 promotes the recruitment and cytotoxicity of NK cells and induces the production of inflammatory cytokines in OSCC cells.

Methods

Study patients

In this study, we recruited 80 primary OSCC patients admitted at the Shandong University Qilu Hospital between February 2014 and December 2016. All the patients underwent surgical treatment and were histopathologically diagnosed. We collected the following data from all the patients: age, gender, and clinicopathological characteristics, including tumor site and TNM stage (according to the AJCC 7th edition). There were a total of 55 male and 25 female patients, and the median age at diagnosis was 63 y (31–82 y). The tongue was the most commonly affected oral site, followed by the gingiva. Among the 80 cases of OSCC, 34 were highly differentiated, 38 were moderately differentiated, and four were poorly differentiated and four were censored. In addition, lymph node metastasis was detected in 24 of the 80 patients. According to the TNM classification, 21, 28, 19, and 12 patients were classified into stages I, II, III, and IV, respectively. The detailed baseline levels of the clinicopathological characteristics are summarized in Table 1. Furthermore, we tested the expressions of A2BR, PD-L1, and CD56 in the tissue samples obtained from the patients.

Table 1 Baseline clinic-pathological characteristics.

Charcteristics	N (%)	
Age (Median, Range)	63 (31–82)	
>63	35 (43.8)	
≤ 63	45 (56.2)	
Sex		
Male	55 (68.8)	
Female	25 (31.2)	
Smoking		
Current smoker	36 (45)	
Current reformed smoker	6 (7.5)	
Non-smoker	38 (47.5)	
AlcoholConsumption		
Yes	42 (52.5)	
No	38 (47.5)	
TumorSite		
Buccal mucosa	6 (7.5)	
Floor of mouth	5 (6.2)	
Gum	13 (16.2)	
Lip	8 (10)	
Palate	7 (8.8)	
Tongue	25 (31.2)	
Others	16 (20)	
Differentiation		
Medium	38 (47.5)	
High	34 (42.5)	
Low	4 (5.0 )	
censored	4 (5.0)	
TumorSize		
≤ 2	31 (38.8)	
2–4	34 (42.5)	
>4	15 (18.8)	
LymphNode Metastasis		
Positive	24 (30)	
Negative	56 (70)	
TNMStage		
I	21 (26.2)	
II	28 (35)	
III	19 (23.8)	
IV	12 (15)	

Ethics statement

All the patients gave written informed consent to participate in these trials according to the Declaration of Helsinki. All the experiments in this study were approved by the Ethics Committee of Shandong University Qilu Hospital (KYLL-2017(KS)-281).

Cell lines

The human OSCC cell line, CAL27, was purchased from ATCC (Gaithersburg, MD, USA), and the cells were cultured in Dulbecco’s modified eagle medium containing 10% fetal bovine serum (FBS, Gibco-Invitrogen, Carlsbad, CA, USA), 100 U/mL of penicillin, and 100 µg/mL of streptomycin. The human NK cell line, NK92, was provided by the Second Military Medical University (Shanghai, China), and the cells were cultured in a an NK92-specific media (Procell, Wuhan, China). The primary NK cells were harvested from the peripheral blood mononuclear cells of a healthy donor (26 y, female) on March 3, 2019, using sucrose density gradient centrifugation. The NK cells were centrifuged at 1174 g at room temperature for 30 min, and the CD56+ cells were sorted using magnetic beads (Miltenyi, Bergisch Gladbach, Germany) according to the manufacturer’s instructions. The purified NK cells were cultured in RPMI1640 media (Gibco-Invitrogen, Carlsbad, CA, USA) containing 10% FBS, 100 U/mL of penicillin, and 100 µg/mL of streptomycin. All the cells were maintained in a humidified incubator at 37 °C and 5% CO2.

Immunohistochemistry (IHC) staining

The OSCC tissue samples were first paraffin-embedded and sectioned into 5 µm thickness and then subjected to IHC staining. The sections were then incubated overnight with anti-A2BR (1:200; Bioss), anti-PD-L1 (1:500; Servicebio, China), or anti-CD56 (1:300; Servicebio, Wuhan, China) antibodies at 4 °C. Thereafter, the tissues were stained with 3′-diaminobenzidine (DAB) solution and counterstained with hematoxylin for nuclei staining. To confirm staining specificity, triplicate sections were immunostained without exposure to primary antibodies (negative control). The chromogenic IHC-stained slides were scanned using the bright field protocol and photographed using the Vectra 3.0 spectral imaging system (PerkinElmer). The images were then analyzed using the InForm v.2.2.1 image analysis software (PerkinElmer). The staining densities of the A2BR and PD-L1 proteins were quantified and graded as low, medium, or high, with a score of 1, 2, or 3, respectively. Additionally, the number of positively-stained cells was counted and graded as zero, no immunostaining; low, <25% positive cells; medium, 26%–50% positive cells; or high, >50% positive cells with a score of 0, 1, 2, or 3, respectively. The expression index for each sample was calculated by multiplying the staining density score with the cell count score. The expression was regarded as weak if the expression index was < 4 and strong if the expression index was ≥ 4. CD56+ cells were counted in five microscopic fields at 400x magnification. A2BR-, PD-L1-, and CD56+-positive cells were scored in cancer nests, margins, and stroma regions of the OSCC tissues. Cancer nests were defined as cancer tissues without fibroblasts and vasculatures, cancer stroma was defined as connective tissues surrounding cancer nests without any cancer cells, and margins were defined as the zone between the outer edge of the cancer nests and the edge of the tissue.

Co-inhibition experiments

CD274 (PD-L1-neutralizing antibody) was purchased from BioLegend (USA), BAY60-6583 (A2BR agonist) and MRS-1706 (A2BR inverse agonist) were purchased from MedChemExpress (USA), and PDTC (NF- κB inhibitor) was purchased from Abcam (UK). CAL27 cells were seeded at a density of 1 ×105/mL in 6 well plate (Corning, Corning, NY, USA) and treated with 100 nM, 1 µM, or 10 µM BAY60-6583 for 24 h or treated with 4 µg/mL PDTC 1 h before BAY60-6583 treatment. Additionally, CAL27 cells were seeded at a density of 1 ×105/mL and treated with 4 µg/mL CD274 and/or 1 µM MRS-1706 for 24 h, as the control group, MRS-1706 group, PD-L1 Nab group and PD-L1 Nab+MRS group. The media was changed and supernatants were collected after 24 h of incubation.

In vitro migration assay

The supernatants were placed in the lower chamber and 100 μL of NK cells (1 ×106/mL) were placed in the upper chamber of a 24-well transwell plate. The cells were resuspended in serum-free media (Corning, USA) and maintained at 37 °C and 5% CO2 in a humidified incubator for 8 h. The migrated cells were fixed with 10% of formalin, stained with crystal violet, and observed under a microscope (Olympus Corporation, Tokyo, Japan). The cells were counted in five randomly selected fields at 400x magnification.

Real-time quantitative reverse transcription PCR (qRT-PCR) assay

Total mRNA was extracted from the cells using TRIzol reagent (Thermo Fisher Scientific, USA), and cDNA was synthesized using a reverse transcription kit (Beijing Transgen Biotech, China). The qRT-PCR assay was performed using Power SYBR Green Master Mix (Takara, Kusatsu, Japan) on a StepOnePlus Real-Time PCR System (Applied Biosystems, Thermo Fisher Scientific, Waltham, MA, USA). Glyceraldehyde 3-phosphate dehydrogenase (GAPDH) was used as an internal control. The following primers were used for the qRT-PCR assay—PD-L1: forward primer 5′-TTCCCAGTCCAAACTGAGGAGTCCAAC-3′ and reverse primer 5′-TTGTTCGCTACCCGAAACGCTGAG-3′ and GAPDH: forward primer 5′-CCAGGTGGTCTCCTCTGACTT-3′ and reverse primer 5′-GTTGCTGTAGCCAAATTCGTTGT-3′. The expression of the target gene was calculated using the 2−ΔΔCT method. Each experiment was replicated thrice and the mean value was obtained.

Flow cytometry assay

The CAL27 cells were added to flow cytometry tubes along with 2 mL of fluorescence-activated cell sorting (FACS) buffer and centrifuged at 400 g for 10 min to wash the cells. The cells were then treated with 100 µL Fc receptor blocking buffer for 15 min at 4 °C. Thereafter, the cells were incubated with 2 mL of FACS buffer and mouse anti-human PD-L1 phycoerythrin (PE)-conjugated monoclonal antibody (R&D systems, USA) or mouse lgG2A PE-conjugated monoclonal antibody (control) (R&D systems; USA) in the dark for 30 min. The molecular expression levels of PD-L1 on the surface of the cells were measured using a flow cytometer (Guava Flow Cell for easyCyte systems; Millipore, Burlington, MA, USA), and the GuavaSoft analysis software (Millipore) was used for data analysis. Apoptotic levels of the CAL27 cells were evaluated using an apoptosis test kit (Xinbosheng, China) according to the manufacturer’s instructions.

Western blot assay

Equal amounts of total protein (50 µg) from each lysate were subjected to 10% sodium dodecyl sulfate–polyacrylamide gel electrophoresis and then transferred to polyvinylidene fluoride membranes. The membranes were blocked with 5% skim milk in Tris-buffered saline containing Tween 20 and incubated overnight with anti-phospho-NF-κBp65 (Ser536) antibodies (1:1000; Cell Signaling Technology, Danvers, MA, USA) and anti-NF-κBp65 antibodies (1:1000; Abcam, Cambridge, UK) at 4 °C. The membranes were washed thrice with wash buffer (Beyotime Biotechnology, Jiangsu, China) and incubated with horseradish peroxidase-conjugated goat anti-rabbit secondary antibodies (1:1000; Beyotime Biotechnology, China) for 1 h at room temperature. Lastly, the membranes were subjected to DAB staining to detect the protein bands.

Statistical analysis

Quantitative variables were evaluated using the student’s t-test or Mann–Whitney test according to their distribution. Two groups of categorical variables were evaluated using the chi-squared test. Quantitative variables, such as age and tumor size were converted to categorical variables (age: ≤63 y and > 63 y, based on the median value, 63 y; tumor size: ≤2 cm, 2–4 cm, and > 4 cm). The in vitro data was analyzed by one-way analysis of variance (ANOVA) and Bonferroni’s test (*p < 0.05; **p < 0.01; ***p < 0.001). Statistical analysis and plotting were performed using GraphPad Prism v.7 (GraphPad, San Diego, CA, USA).

Results

Differential expression pattern of A2BR and PD-L1 in OSCC tissues

To examine the expression pattern of A2BR and PD-L1 in OSCCs, we performed IHC staining of tissue samples obtained from 80 OSCC patients. Additionally, we analyzed the expression intensity of A2BR and PD-L1 in the cancer nest, border, and paracancer stroma regions of the tissue samples. As shown in Fig. 1A, PD-L1 showed the highest expression in the border regions (48.53%), followed by paracancer stroma (40.00%), and cancer nest (12.68%), while A2BR showed the highest expression in the cancer nest (34.25%), followed by border (19.72%), and paracancer stroma (3.23%; Figs. 1B and 1C). We further conducted a correlation analysis between the expressions of A2BR and PD-L1 in the tumor nests and border regions, since the sample size of A2BR expression in paracancer stroma was insufficient. As shown in Fig. 1D, the expression intensities of A2BR and PD-L1 were not correlated in the tumor nest (p = 0.458); however, most of the samples that showed high A2BR expression in the border regions, also exhibited high PD-L1 expression (13/14, 93%, p < 0.01). Since the border region is characterized by the most active tumor growth, we hypothesized that the correlation between A2BR and PD-L1 expression may be associated with OSCC progression.

Figure 1 Differential expression pattern of adenosine 2b receptor (A2BR) and programmed death-ligand 1 (PD-L1) in oral squamous cell carcinoma (OSCC) tissues.

(A) Representative images of PD-L1 and A2BR expression in the tumor nest, border, and paracancer stroma regions (100x and 400x magnification) of the OSCC tissue samples obtained from 80 OSCC patients. (B) The expression indexes of PD-L1 and A2BR in OSCC samples. (C) The expression was regarded as weak if the expression index was <4 and strong if the expression index was ≥ 4. The proportion of OSCC samples showing strong/weak PD-L1 and A2BR expression. (D) The association between PD-L1 and A2BR expression in the OSCC samples. Statistical analysis was performed using the chi-squared test.

A2BR promoted PD-L1 expression via the NF-κB pathway in OSCC cells

We performed in vitro experiments to examine the role of A2BR in PD-L1 expression and the results revealed that activation of A2BR by BAY60-6583 enhanced the transcription (Fig. 2A) and cell surface localization of PD-L1 (Figs. 2B and 2C) in CAL27 cells. Furthermore, BAY60-6583 treatment significantly increased NF-κBp65 phosphorylation, suggesting that stimulation of A2BR can lead to the activation of the NF-κB signaling pathway (Fig. 2D). However, compared with BAY60-6583 treatment, pretreatment with PDTC, 1 h prior to BAY60-6583 treatment, partially reduced PD-L1 expression (Fig. 2E). Altogether, these results demonstrate that A2BR promotes the expression and cell surface localization of PD-L1 via the NF-κB signaling pathway.

Figure 2 Adenosine 2b receptor (A2BR) promoted programmed death-ligand 1 (PD-L1) expression via the nuclear transcription factor-kappa B (NF-κ B) pathway in oral squamous cell carcinoma (OSCC) cells.

CAL27 cells were treated with BAY60-6583 (A2BR agonist) for 24 h. (A) Real-time quantitative reverse transcription PCR assay showing the mRNA levels of PD-L1 in CAL27 cells after BAY 60-6583 treatment. (B) Flow cytometry assay showing the surface localization of PD-L1 in CAL27 cells after BAY60-6583 treatment. (C) The mean fluorescence intensity of PD-L1 in CAL27 cells after BAY60-6583 treatment. (D) Western blot assay showing the levels of pNF-κ Bp65 after BAY60-6583 treatment. (E) Flow cytometry assay showing the surface localization of PD-L1 after pretreatment with PDTC (NF-κ Bp65 inhibitor) 1 h prior to BAY60-6583 treatment. Paired Student’s t-test was used for statistical analysis. The results are represented as mean ± SD. * P < 0.05,** P < 0.01 and ** P < 0.01.

A2BR and PD-L1 expression was associated with NK cell infiltration in OSCC tissues

We further examined the association between the expression intensities of A2BR and PD-L1 with NK cell infiltration in the cancer nest, border, and stroma regions of the OSCC samples. The results revealed that the number of CD56+ NK cells was the highest in tumor nest (9,531 ± 1,183), followed by border (3,029 ± 413.9), and stroma (1,325 ± 198.6; p < 0.05; Fig. 3A). We further found that the average numbers of NK cells in the tumor nest and border regions were higher in samples exhibiting strong expression of both A2BR and PD-L1 compared to those exhibiting weak expression of A2BR and PD-L1 (Fig. 3B), although the results were not significant, due to the high variation in the sample numbers between the two groups. However, we found that NK cell infiltration in the tumor nest regions was significantly lower in the high-A2BR expression samples than in the low-A2BR expression samples (median values: 5,302 and 10,751, respectively, p = 0.003), while there was no significant difference in the number of NK cells in the border regions between the high- and low-A2BR expression samples (median value: 2,538 and 1,054, respectively, p = 0.176; Fig. 3C).

Figure 3 Adenosine 2b receptor (A2BR) and programmed death-ligand 1 (PD-L1) expression was associated with natural killer (NK) cell infiltration in oral squamous cell carcinoma (OSCC) tissues.

(A) The number of CD56+ NK cells in tumor samples obtained from 80 OSCC patients. (B) The number of CD56+ NK cells in OSCC samples showing strong/weak PD-L1 and A2BR expression. (C) The number of infiltrated CD56+ NK cells in the tumor nest and border regions of the tissue samples showing strong/weak PD-L1 (Left) and A2BR (Right) expression after stratification according to the expression index. Mann–Whitney test was used for statistical analysis. ** P < 0.01.

Treatment with MRS-1706 and CD274 synergistically increased NK cell recruitment in OSCC cells

To explore whether A2BR serves as a target for anti-PD-L1 immunotherapy, we treated CAL27 cells with MRS-1706 and/or CD274 and harvested the supernatants to examine their effects on NK cell recruitment and pro-inflammatory cytokine release. The results suggested that CD274-treated samples showed increased NK cell recruitment compared to the untreated control samples, while MRS-1706-treated samples showed significantly increased NK cell recruitment compared to the CD274-treated samples. However, samples treated with both CD274 and MRS-1706 exhibited relatively higher NK cell recruitment compared to those treated with MRS-1706 alone (Fig. 4). These results suggest that MRS-1706 and CD274 synergistically increase NK cell recruitment in OSCC.

Figure 4 Treatment with MRS-1706 and CD274 synergistically increased natural killer (NK) cell recruitment in oral squamous cell carcinoma (OSCC) cells.

CAL27 cells were stimulated with CD274 (PD-L1 neutralizing antibody) and/or MRS-1706 (A2BR inverse agonist) for 24 h. The media was changed and the supernatants were obtained after 24 of incubation. The supernatants were placed in the lower compartment and the NK cells were placed in the upper compartment of a transwell plate and NK cell migration was observed. (A) Representative images of control group, MRS-1706 group, PD-L1 Nab group and PD-L1 Nab+MRS group (40x magnification). (B) The number of migrated NK cells in the four treatment groups. Paired Student’s t-test was used for statistical analysis. The results are represented as mean ± SD. * P < 0.05 and ** P < 0.01.

Treatment with MRS-1706 and CD274 synergistically promoted NK cell cytotoxicity against OSCC cells

To explore whether A2BR affects NK cell cytotoxicity against tumor cells, we established a CAL27 and NK92 cell co-culture system and treated it with MRS-1706 and/or CD274. The results revealed that the apoptotic level of MRS-1706-treated CAL27 cells was higher than that of the untreated cells, while the apoptotic level of CD274-treated cells was higher than that of the MRS-1706-treated cells (Fig. 5A). However, the apoptotic level of the cells treated with both MRS-1706 and CD274 was higher than those of the cells treated with MRS-1706 or CD274 alone (Fig. 5B). These results suggest that MRS-1706 and CD274 synergistically increase NK cell cytotoxicity.

Figure 5 Treatment with MRS-1706 and CD274 synergistically promoted natural killer (NK) cell cytotoxicity against oral squamous cell carcinoma (OSCC) cells.

(A) Representative images of the apoptotic level of the CAL27 cells co-cultured with NK92 and treated with CD274 (PD-L1 neutralizing antibody) and/or MRS-1706 A2BR inverse agonist), as detected by flow cytometry assay. (B) The percentage of apoptotic CAL27 cells co-cultured with NK cells in the control group, MRS-1706 group, PD-L1 Nab group and PD-L1 Nab+MRS group . Paired Student’s t-test was used for statistical analysis. The results are represented as mean ± SD. * P < 0.05 and ** P < 0.01.

Discussion

In this study, we obtained tumor tissue samples from 80 OSCC patients admitted at the Shandong University Qilu Hospital between February 2014 and December 2016. Gene expression analysis revealed that A2BR was highly expressed in the tumor nests and PD-L1 was highly expressed in the border regions of these samples. Interestingly, we found that most of the samples exhibiting strong A2BR expression in the border regions also exhibited strong PD-L1 expression, suggesting that A2BR and PD-L1 expression is correlated in the border regions. Furthermore, we found that treatment with BAY60-6583, an A2BR agonist, promoted PD-L1 transcription and cell surface localization in the CAL27 cell line, indicating that A2BR activation stimulates PD-L1 expression. Since PD-L1 is an important target for immunotherapy, our study provides a novel PD-L1-targeting strategy for anti-cancer treatments.

Considering the importance of PD-L1 in tumor immunity, it is necessary to elucidate the mechanism by which A2BR regulates PD-L1 expression. NF-κB is an important transcription factor, involved in the regulation of various genes associated with anti-apoptosis, inflammation, and oncogenesis (Kaur et al., 2023; Peng et al., 2020). Overexpression of NF-κB and activation of the NF-κB signaling pathway is associated with immune responses in several diseases (Capece et al., 2022). Increasing evidence suggests that the NF-κB pathway is activated in the etiology and development of head and neck squamous cell carcinomas (Liao et al., 2020; Morgan, Chen & Van Waes, 2020; Cui et al., 2022). Our results showed that BAY60-6583 significantly increased the phosphorylation level of NF-κBp65, suggesting that A2BR stimulation can activate the NF-κB signaling pathway. These results are consistent with the results of the previous studies which found that adenosine receptors lead to disease progression by regulating the NF-κB signaling pathway (Zhou et al., 2022). Moreover, we found that pretreatment with PDTC, an NF-κB inhibitor, partially diminished the enhancing effects of BAY60-6583 on PD-L1 expression, suggesting that A2BR promotes the PD-L1 expression in OSCC via the activation of the NF-κB signaling pathway.

Antagonizing A1, A3, or A2A receptors promotes the infiltration of inflammatory cells, such as monocytes and macrophages, in adenosine-mediated vasculitis and plays critical roles in TME (Signa et al., 2022). However, in sepsis patients, activation of the A2A receptor inhibits the activation of the NF-κB pathway and alleviates inflammatory cell infiltration (Sun, Wang & Hao, 2022). Moreover, since NF-κB pathway is closely associated with immune response (Pflug & Sitcheran, 2020), it plays a significant role in PD-L1 expression.

We further found that the number of CD56+ NK cells was the highest in the tumor nest regions compared to the border and stroma regions of the OSCC tissue samples, indicating that CD56+ NK cells were primarily recruited to the tumor nest, where the tumor cells predominantly accumulate. These results are consistent with the cytotoxic function of the CD56+ NK cells. Tumor nests with high A2BR expression showed lower NK cell infiltration than those with low A2BR expression, suggesting that NK cell infiltration was negatively associated with A2BR expression in tumor nests. However, there was no correlation between A2BR expression and NK cell infiltration in the border regions of the OSCC samples. This may be due to the reduced quantity of tumor cells in the border region compared to that in the tumor nest. Several reports showed that the PD-1/PD-L1 pathway directly and indirectly regulates NK cells in cancers (Thacker et al., 2023; Munari et al., 2021). For instance, studies found that PD-1/PD-L1 is associated with NK cell infiltration in lymphoma (Rong et al., 2021) and HNC (Meliante et al., 2022) and that it regulates anti-tumor immunity. However, in this study, we found that PD-L1 expression was not significantly associated with NK cell infiltration in both tumor nests and border regions, suggesting that PD-L1 may be not a high-efficiency regulator of NK cell infiltration in OSCC. To further confirm the above findings, we conducted a series of in vitro experiments and found that antagonizing A2BR enhanced the infiltration and cytotoxicity of NK cells in the CAL27 cell line. In addition, CD274-mediated neutralization of PD-L1 further enhanced the promotive effects of MRS-1706-mediated antagonization of A2BR on NK cell infiltration and cytotoxicity; therefore, A2BR antagonization along with PD-L1 neutralization can be a potential strategy for tumor therapy based on enhancing the NK cell function.

There are some limitations to our study. Firstly, our study was based on in vitro experiments using a single cell line and tissue samples from OSCC patients and further validation is required using more cell lines, animal models, and clinical trials. Secondly, although we demonstrated that the NF-κB signaling pathway plays a regulatory role in PD-L1 expression and NK cell infiltration and function, further studies are required to verify the role of NF-κB signaling pathway in A2BR-mediated regulation of PD-L1 expression and NK cell infiltration and function. Additionally, further studies are required to elucidate the alternative mechanisms associated with A2BR-mediated regulation of PD-L1 expression and NK cell infiltration and function. Thirdly, the sample size of the clinical cohort in this study was relatively small, thus large-scale studies are needed to confirm the correlation between A2BR expression, NK cell infiltration, and the overall survival of OSCC patients. Lastly, the safety and potential side effects of co-targeting A2BR and PD-L1 need to be carefully evaluated, since targeting immune checkpoint pathways can lead to immune-related adverse events.

In this study, we explored the effects and significance of A2BR on PD-L1 expression and NK cell recruitment and cytotoxicity in OSCC. The results of our study provide important insights into the potential therapeutic strategy for OSCC treatment based on the co-inhibition of A2BR and PD-L1.

Supplemental Information

Supplemental Information 1 Uncropped blot.

Click here for additional data file.

Not Applicable

Additional Information and Declarations

Competing Interests

Author Contributions

Human Ethics

Data Deposition

The authors declare that there are no competing interests.

Bing Wang conceived and designed the experiments, performed the experiments, analyzed the data, prepared figures and/or tables, and approved the final draft.

Tao Wang conceived and designed the experiments, performed the experiments, analyzed the data, prepared figures and/or tables, and approved the final draft.

Chengzhe Yang conceived and designed the experiments, authored or reviewed drafts of the article, and approved the final draft.

Zhaodi Nan performed the experiments, authored or reviewed drafts of the article, and approved the final draft.

Dan Ai performed the experiments, prepared figures and/or tables, and approved the final draft.

Xin Wang analyzed the data, prepared figures and/or tables, and approved the final draft.

Huayang Wang conceived and designed the experiments, prepared figures and/or tables, and approved the final draft.

Xun Qu analyzed the data, authored or reviewed drafts of the article, and approved the final draft.

Fengcai Wei conceived and designed the experiments, authored or reviewed drafts of the article, and approved the final draft.

The following information was supplied relating to ethical approvals (i.e., approving body and any reference numbers):

All experiments in this study were approved by the Ethics Committee of Shandong University Qilu Hospital (KYLL-2017(KS)-281).

The following information was supplied regarding data availability:

The data is available at Figshare: Bing, Wang (2023). OSCC data. figshare. Journal contribution. https://doi.org/10.6084/m9.figshare.23266277.v1.

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
