# Peer review of "Co-inhibition of adenosine 2b receptor and programmed death-ligand 1 promotes the recruitment and cytotoxicity of natural killer cells in oral squamous cell carcinoma"

_PeerJ, doi:10.7717/peerj.15922_

## Round 0.1 · original submission · Minor Revisions

All these reviewers had useful comments and suggestions. please revise your manuscript according to these comments.

Reviewer 1 ·

Basic reporting

In the study, the author first discovered the correlation between A2BR and PD-L1 expression in oral paracancerous tissues through a clinical cohort of OSCC patients. Then the author confirmed through a series of in vitro experiments that A2BR agonists can induce PD-L1 expression by inducing the NF-kB pathway, and inversely activate A2BR, and synergize with PD-L1 antibodies to increase the recruitment and cytotoxicity of NK cells. In conclusion, A2BR-targeted inhibitory combined with PD-L1 antibody to enhance the cellular effects of NK cells may be a new tumor immunotherapy strategy. The manuscript is well written, comprehensive and clear to the reader. The results of this study are consistent with the research objects, and the discussion is detailed.The manuscript is recommend to publish in this journal, however, there are few issue should be fully addressed.

Experimental design

In the study, the author first discovered the correlation between A2BR and PD-L1 expression in oral paracancerous tissues through a clinical cohort of OSCC patients. Then the author confirmed through a series of in vitro experiments that A2BR agonists can induce PD-L1 expression by inducing the NF-kB pathway, and inversely activate A2BR, and synergize with PD-L1 antibodies to increase the recruitment and cytotoxicity of NK cells. Generally, the research is orignial and meet the aim and scope of this journal with a rigourous investigation design to a high technical and ethical standard. Methods described with sufficient and detailed information to replicate.

Validity of the findings

In conclusion, A2BR-targeted inhibitory combined with PD-L1 antibody to enhance the cellular effects of NK cells may be a new tumor immunotherapy strategy. Meaningful replication encouraged and all underlying data have been provided. Conclusions are well stated.

Additional comments

Minor concern:
1. The author mentioned the “cancer nests” and “margins” in the article, but did not give a clear definition. It would be more clear if the author could define them in detail in the article
2. In Table 1, the total number of patients divided according to the degree of tumor differentiation was less than 80, which should be explained by the author.
3. In the results section, the author found that in samples with strong expression of both A2BR and PD-L1 in either region, the average numbers of NK cells were higher. However, the author also confirmed that, in the tumor nest, the number of NK cells in samples with strong A2BR expression was lower than that with weak A2BR expression. Why these two situations occurred, the author should have a full discussion.
4. In Fig.1A, the local enlarged images of tumor, border and stroma were not completely consistent with the circled area in the original image, the author should check carefully.
5. In the discussion section, the authors stated that “A2BR enhances the infiltration of NK cells through activation of NF-kB pathway”, but the authors did not prove it in the paper; In addition, the authors stated that “Activation of A2BR, synergistic with PD-
L1 antibody, can increase the recruitment and cytotoxicity of NK cells”, but in this article the author used an A2BR inverse agonist in synergy with the PD-L1 antibody. The author's statements should be more accurate.
6. There are several grammatical and spelling errors in this manuscript, such as “TEM”. Please review the text carefully and correct any errors.

Reviewer 2 ·

Basic reporting

1.This study investigated the protective effect of A2BR and PD-L1 antibody for OSCC. They found A2BR agonists induced PD-L1 expression through induction of NF-kB. High expression of A2BR was association to low NK cells infiltration. Moreover, A2BR inverse agonist synergistic with PD-L1 neutralizing antibody recruited NK cells to induce OSCC cell apoptosis. The research idea is clear and the method is reasonable. However, a large number of language errors make the article poorly readable, such as line 241-243 and the legend of figure 4. The author should review the full text carefully and have the language polished by a professional.
2.For the title, it is generally not recommended to have an abbreviation, in addition, the author only studied apoptosis of cancer cells, so it should be described: against oral squamous cell carcinoma apoptosis.
3.The full names of abbreviations should be provided in the abstract, main text, table notes and legends when they 1st appear. For example, A2BR, PD-L1, NF-kB Please provide their full names. please checked the full text.
4.The description of the abstract follows the background, method, result and conclusion. At present, I can't see what method the author uses to operate.

Experimental design

5.The method should be more detailed, such as how many grams of protein are used for electrophoresis.
6.In the part of statistical methods, what methods are used for testing more than three groups and post-test.
7. In the results section, information about the clinical characteristics of the sample should be placed in the method section.

Validity of the findings

8.In the results section, in most cases, the authors only stated a result, but did not analyze the cause of the result. The authors should combine some literatures and put forward their own views and conjectures. Authors can add few lines of summary in each small part of results section and emphasize what is the reason for this result, and what hints can this result give us?
9. In the Discussion, this part should be re-revised deeply. The discussion is not a replicate of results, delete the repeated contents. The authors should combine current results and previous results to ascertain the validity of current studies and discuss the potential mechanisms or give an explanation of the difference.
10.For statistical p-values, it is generally p. however, P, P all presented in paper, please uniformly it.
11.In figure 4, please define symbol *, **, in figure legend.

Additional comments

12.Are there any limitations to this study? Please add a paragraph to the discussion.

Reviewer 3 ·

Basic reporting

1. Wang et al found that adenosine receptor A2BR is co-expressed with PD-L1 in oral squamous cell carcinoma (OSCC). Using in vitro experiments, they also discovered that A2BR agonists can induce PD-L1 expression through the activation of NF-kB pathway, and that A2BR expression is negatively correlated with the infiltration of NK cells. Importantly, the study demonstrated that the inhibition of both A2BR and PD-L1 can promote the recruitment and cytotoxicity of NK cells against OSCC. These findings suggest that targeting both A2BR and PD-L1 could promote anti-tumor immune responses and improve the efficacy of immune checkpoint therapy in OSCC. This is a topic of interest to researchers in related fields, with clear research objectives, but the professional English of this article needs to be improved.
2. The citation of references is necessary to check, and the literature in the past three years should be cited. In addition, it is necessary to revise the format of the reference list according to PeerJ's specifications.
3. The gray value of NF-KBp65 and pNF-KBp65 needs to be statistically analyzed in Fig 2D. The image of Figure 4A requires adding a ruler.

Experimental design

1. While this study provides important insights into the potential of targeting both A2BR and PD-L1 for the treatment of OSCC, there are some limitations that should be considered. Firstly, this study was based on in vitro experiments using cell lines and supernatants from OSCC patients, and further validation is needed in animal models and clinical trials. Secondly, the mechanism of how A2BR activation promotes PD-L1 expression and the inhibitory effect on NK cell infiltration and function needs to be further elucidated. Thirdly, the sample size of the clinical cohort is relatively small, and larger studies are needed to confirm the correlation between A2BR expression, NK cell infiltration, and overall survival in OSCC patients. Finally, the safety and potential side effects of targeting both A2BR and PD-L1 in combination need to be carefully evaluated, as targeting immune checkpoint pathways can lead to immune-related adverse events.
2. In the Introduction part, the author indicated that “However, less than 20% patients with OSCC benefit from this therapy because of tumor heterogeneity (Ferris et al.2016; Neal et al. 2018).” Only one OSCC cell line was used in this study, which may not represent the heterogeneity of OSCC. Further studies with a larger dataset and multiple OSCC cell lines are required to confirm the effectiveness of A2BR inverse agonist synergistic with PD-L1 in different OSCC subtypes.

Validity of the findings

1. In the Results part, the author indicated that “The detailed baseline levels of clinicopathological characteristics were summarized in Table 1.” The key clinicopathological characteristics should be described in the manuscript.
2. Some of the descriptions in the discussion were inappropriate, and the author needs to make revisions. such as “Interestingly, we found that in tissue samples with strong A2BR expression in the border areas, PD-L1 also exhibits strong expression in most samples, suggesting that the local expression of A2BR and PD-L1 is corelated to certain extent.” “In tissue samples of oral cancer, we further found that the number CD56+NK cells is the highest in the tumor nest, and where the number of NK cells strongly expressing A2BR is lower than that of NK cells weakly expressing A2BR, while the expression of PD-L1 is not significantly associated with the infiltration of NK cells. These findings suggest that, the NK cell infiltration in the tumor nest in OSCC might be related to A2BR.”.
3. The proportion of expression index of PD-L1 and A2BR in various tissues was presented in Fig 1B. The author should verify the consistency of the sample size before and after, including Figure 1D.

Additional comments

no comment

---

## Round 0.2 · accepted · Accept

In carefully evaluating the contents of this revised paper, I was satisfied with the responses and revisions made by the authors. The Reviewer's concerns have been well addressed. With the necessary revisions and improvements, the quality of this paper has been significantly improved. I believe that this revised manuscript is ready to be considered for publication in this journal.